# An intranasal influenza virus-vectored vaccine prevents SARS-CoV-2 replication in respiratory tissues of mice and hamsters

Shaofeng Deng [1,2], Ying Liu[1,2], Rachel Chun-Yee Tam[1,2], Pin Chen[1,2], Anna Jinxia Zhang [1,2,3], Bobo Wing-Yee Mok[1,2,3], Teng Long [1,2,3], Anja Kukic[1,2], Runhong Zhou[1,2], Haoran Xu[1,2], Wenjun Song [1,2], Jasper Fuk-Woo Chan [1,2,3], Kelvin Kai-Wang To [1,2,3], Zhiwei Chen [1,2,3], Kwok-Yung Yuen [1,2,3], Pui Wang [1,2,3] ✉ & Honglin Chen [1,2,3] ✉

Current available vaccines for COVID-19 are effective in reducing severe diseases and deaths caused by SARS-CoV-2 infection but less optimal in preventing infection. Next-generation vaccines which are able to induce mucosal immunity in the upper respiratory to prevent or reduce infections caused by highly transmissible variants of SARS-CoV-2 are urgently needed. We have developed an intranasal vaccine candidate based on a live attenuated influenza virus (LAIV) with a deleted NS1 gene that encodes cell surface expression of the receptor-binding-domain (RBD) of the SARS-CoV-2 spike protein, designated DelNS1-RBD4N-DAF. Immune responses and protection against virus challenge following intranasal administration of DelNS1-RBD4N-DAF vaccines were analyzed in mice and compared with intramuscular injection of the BioNTech BNT162b2 mRNA vaccine in hamsters. DelNS1-RBD4N-DAF LAIVs induced high levels of neutralizing antibodies against various SARS-CoV-2 variants in mice and hamsters and stimulated robust T cell responses in mice. Notably, vaccination with DelNS1-RBD4N-DAF LAIVs, but not BNT162b2 mRNA, prevented replication of SARS-CoV-2 variants, including Delta and Omicron BA.2, in the respiratory tissues of animals. The DelNS1-RBD4N-DAF LAIV system warrants further evaluation in humans for the control of SARS-CoV-2 transmission and, more significantly, for creating dual function vaccines against both influenza and COVID-19 for use in annual vaccination strategies.

The late 2019 emergence of SARS-CoV-2 in humans led to the COVID-19 pandemic, which has had an unprecedented global impact on human health, social stability, and the economy[1,2]. Rapid development and deployment of several vaccines for emergency use in humans has significantly alleviated COVID-19 disease caused by infection with SARS-CoV-2[3,4]. However, continuous circulation of SARS-CoV-2 in humans has led to the emergence of highly transmissible variants of concern (VOC) with immune evasion abilities that pose a considerable challenge to efforts to bring the COVID-19 pandemic to an end[5–8]. While variants exhibit differing pathogenic properties, with the most

[1]Department of Microbiology, Li Ka Shing Faculty of Medicine, the University of Hong Kong, Pokfulam, Hong Kong Special Administrative Region, People's Republic of China. [2]State Key Laboratory for Emerging Infectious Diseases, the University of Hong Kong, Pokfulam, Hong Kong Special Administrative Region, People's Republic of China. [3]Centre for Virology, Vaccinology and Therapeutics Limited, the University of Hong Kong, Pokfulam, Hong Kong Special Administrative Region, People's Republic of China. ✉e-mail: puiwang@hku.hk; hlchen@hku.hk

recent Omicron variant mainly causing mild disease[9], the combination of high transmission rates and reinfections due to immune evasion can still lead to substantial numbers of hospitalizations and deaths, especially among the elderly[10]. Current mRNA, inactivated whole viral particle and adenoviral vector vaccines are administered intramuscularly and able to induce significant levels of serum antibodies and circulating T cell responses in humans, helping to prevent severe COVID-19 disease or death. Less clear is the level and scope of immunity induced by current COVID-19 vaccines in the upper respiratory tract, where SARS-CoV-2 infection is initiated. The relatively common occurrence of breakthrough infections in vaccinees and repeated SARS-CoV-2 infections in some individuals suggest that alternative vaccine approaches that enhance immunity in the upper airways may be necessary. Enhancement of vaccine-induced immunity in the upper respiratory tract is a key focus in the development of next phase vaccines to reduce SARS-CoV-2 circulation to a low level. Indeed, several intranasally administered vaccine candidates have shown distinct advantages over intramuscularly injected vaccines in inducing mucosal immunity and blocking virus replication in the airways[11–15]. It is highly likely that SARS-CoV-2 will continue to co-circulate with seasonal influenza to cause annual epidemics. A dual function vaccine to prevent both COVID and influenza may be more acceptable to the general population and cost effective for the post pandemic control of epidemics.

We previously reported a panel of live attenuated influenza viruses (LAIV) in which the NS1 gene is deleted from the viral genome (DelNS1) and adaptive mutations support viral replication in embryonated chicken eggs and MDCK cells[16,17]. We have shown immunization with DelNS1 LAIVs to provide cross-protective, and potentially long lasting, immunity in an influenza infection model[16]. We further developed the DelNS1 LAIV system, inserting the receptor-binding domain (RBD) of SARS-CoV-2 at the site of NS1 deletion, and showed that DelNS1-RBD LAIV prime-boost immunization induced strong systemic and mucosal immune responses to block SARS-CoV-2 infection in a mouse model[18]. Phase I/II clinical trials with an early version of the vaccine candidate, DelNS1-nCoV-RBD-OPT1 have shown DelNS1-RBD to be well tolerated[19]. Antibody and T cell responses to the RBD of SARS-CoV-2 were detected among some subjects but we were unable to fully estimate specific immune responses in the airways. Supported by the safety features and promising preclinical evidence, phase III trials to evaluate this vaccine candidate's potential contribution to current intramuscular SARS-CoV-2 vaccine strategies are currently underway in several countries (http://www.chictr.org.cn/showproj.aspx?proj=133897)[20]. To further enhance the immunogenicity of the DelNS1-RBD LAIV and address concerns regarding emerging variants, this study presents data on an improved version of the DelNS1-RBD vaccine candidate, tested against current variants in animal models. Our result shows that DelNS1-RBD4N-DAF LAIVs, which facilitate expression of RBD on the cell surface, are highly immunogenic in terms of inducing neutralizing antibodies against the original SARS-CoV-2 strain and current variants, including Omicron BA.2. Notably, intranasal vaccination with DelNS1-RBD4N-DAF LAIVs, but not intramuscular injection of an mRNA vaccine, can block virus replication in the lungs and nasal turbinates of subsequently infected hamsters. This influenza vector vaccine has the potential to serve as a stand-alone prime-boost or supplementary booster immunization to control SARS-CoV-2 circulation and may be developed into a dual function vaccine for both COVID-19 and influenza.

## Results

### Surface expression and targeted N-glycosylation enhance immunogenicity of SARS-CoV-2 spike RBD in NS1-deleted influenza viral vector system

Non-structural protein 1 (NS1) of influenza virus plays critical roles in the viral life cycle[21,22]. We previously reported an attenuated influenza viral vector system which can be used to express the receptor-binding domain (RBD) of SARS-CoV-2 from the site of NS1 deletion (DelNS1-RBD)[16–18]. When administered intranasally this vaccine candidate was shown to effectively boost systemic and mucosal immune responses in animals. The current study further modifies this system through addition of the gene for a membrane anchored protein, decay accelerating factor (DAF)[23], to enhance expression of RBD on cellular surfaces (Fig. 1A). Fusing SARS-CoV-2 RBD to a short peptide spanning the transmembrane and cytoplasmic domains of DAF ensures that it will be properly processed and presented on the cell surface of DelNS1-RBD-DAF LAIV infected cells. As shown in infected MDCK cells, viral vector expressing the RBD-DAF fusion protein resulted in high levels of membrane displayed RBD, compared to the vector encoding RBD without DAF (Fig. 1B). To test if inclusion of DAF indeed enhances immune responses to RBD, we immunized mice with DelNS1 vector, DelNS1-RBD or DelNS1-RBD-DAF LAIVs using a prime-boost scheme (Fig. 1D) and measured total antibodies to RBD and neutralization activity against a pseudovirus expressing the spike protein of SARS-CoV-2 six weeks after the prime immunization. Inclusion of DAF significantly increased total anti-RBD and neutralizing antibody levels in mice (Fig. 1E). Also critical is the induction of highly specific immunity to the epitopes located within the core ACE2 receptor-binding region of the SARS-CoV-2 RBD. Glycosylation of spike is known to shield viral antigens and may affect exposure of spike protein epitopes for recognition by immune cells[24]. We reasoned that by shielding non-ACE2 competing epitopes outside the receptor-binding motif (RBM), epitopes within the RBM may be more prominently presented. To test this hypothesis, we constructed an RBD with substitution of four residues outside the RBM (A372T, G413N, D428N and P521N), targeting them for N-glycosylation (4 N) (Fig. 1A). Western blot confirmed an increase in the molecular weight of RBDs where residues were substituted to promote N-glycosylation individually or in combination (RBD4N) (Fig. 1C, upper panel). Intriguingly, glycosylation of these residues seems to increase expression of RBD in cell lines representing a range of mammalian species (Fig. 1C, bottom panel). We then examined immune responses to RBD and RBD4N derived from Beta (B.1.351) or Delta (B.1.617.2) variants of SARS-CoV-2 using the DelNS1-RBD-DAF LAIV system in mice. Mice were intranasally immunized with DelNS1 vector or Beta or Delta versions of either DelNS1-RBD4N-DAF or DelNS1-RBD-DAF LAIVs (Fig. 1D) and total antibodies to RBD (WT, lineage A virus)[25] and neutralizing activity measured using surrogate ELISA and pseudovirus assays, respectively. 4N versions of both Beta and Delta RBDs (Beta4N, Delta4N) induced higher levels of antibodies than RBDs lacking 4N modifications (Fig. 1F). Neutralization assays showed that for both Beta and Delta RBDs, additional N-glycosylation enhanced neutralizing antibody responses against pseudoviruses expressing Beta, Delta or Omicron BA.1 variant spike proteins (Fig. 1F). These data demonstrate that fusion with a membrane anchored protein, DAF, to facilitate surface expression of RBD and targeting for additional N-glycosylation to shield non-ACE2 competing epitopes enhances the immunogenicity of RBD in this influenza virus-vectored vaccine system.

### Immunogenicity of DelNS1-RBD4N-DAF variants

Since Omicron subvariants became the major circulating strains in early 2022, replacing Delta and earlier variants, the immunogenicity of DelNS1-RBD4N-DAF vaccine candidates encoding expression of Beta, Delta or Omicron (BA.1) RBDs (Beta4N-DAF, Delta4N-DAF or Omi4N-DAF, respectively) was assessed and compared with that of the BioNTech mRNA vaccine in animals (Fig. 2A), using a prime-boost regimen as described above (Fig. 1D). Both Delta and Omicron (BA.1) DelNS1-RBD4N-DAF LAIV vaccine candidates induced high levels of antibodies specific for the RBD of the original SARS-CoV-2 strain in mice but neutralization assays using pseudoviruses or live viruses showed that each RBD induced more strain-specific antibodies, particularly the

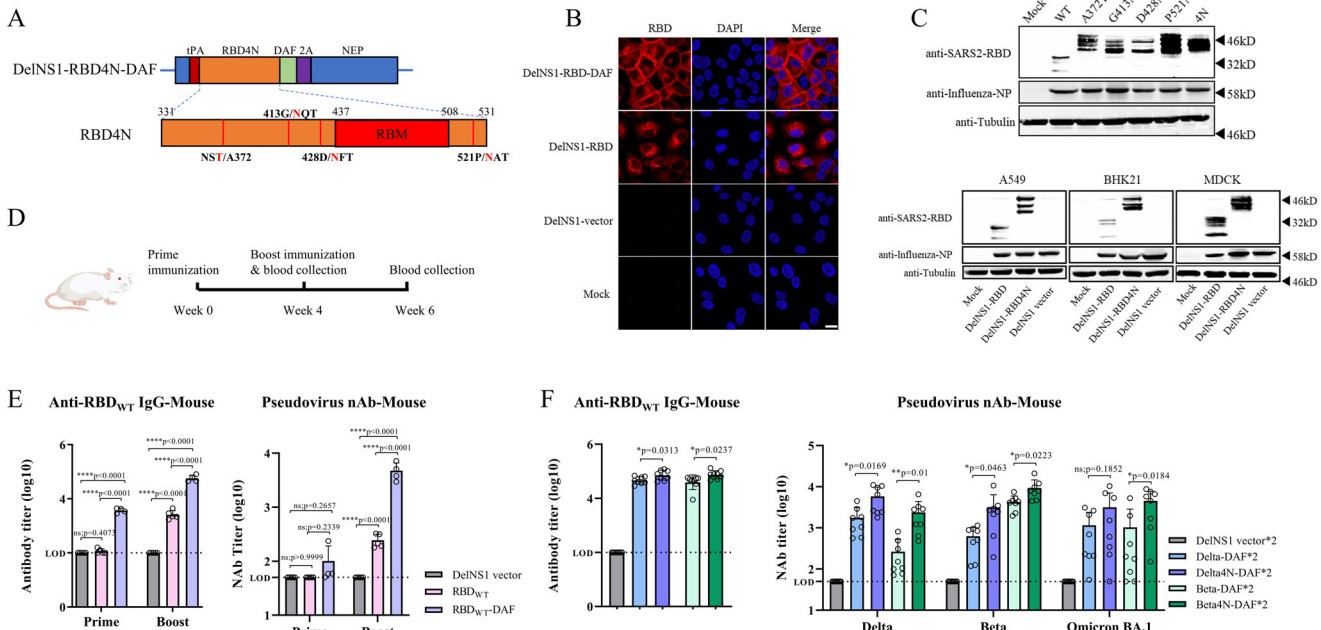

**Fig. 1 | Construction and evaluation of DelNS1-RBD4N-DAF live attenuated virus vaccine candidates. A** Illustration of construction of DelNS1-RBD4N-DAF. **B** Immunofluorescence (IF) assay of RBD expression with or without DAF in MDCK cells. MDCK cells were mock infected or infected with DelNS1, DelNS1-RBD or DelNS1-RBD-DAF viruses in the backbone of A/California/4/2009 (H1N1)[16] at an MOI of 1 for 10 h and cells processed for IFA using antibodies specific for RBD (red). Nuclei were stained with DAPI (blue). Images are representative of three independent experiments. **C** Confirmation of N-glycosylation in RBD of DelNS1-RBD4N-DAF LAIV. MDCK cells were mock infected or infected with DelNS1-RBD (WT, lineage A virus), DelNS1-RBD with individual N-glycosylation mutations (A372T, G413N, D428N, or P521N), or DelNS1-RBD-4N (4N) LAIV virus at an MOI of 1 for 10 h. Cell lysates were analyzed by western blot using anti-RBD antibody and anti-NP antibody. DelNS1-RBD, DelNS1-RBD4N, and DelNS1 LAIVs were also used to infect A549, BHK21 and MDCK cells to detect the expression of RBD. Images are representative of three independent experiments. **D** Schedule of immunization and blood collection for BALB/c mice. **E** Estimation of anti-RBD antibodies in mice prime-boost

immunized intranasally with $2 \times 10^6$ pfu of DelNS1-RBD-DAF with wild-type (WT) SARS-CoV-2 strain (lineage A) RBD ($RBD_{WT}$-DAF, $n = 4$) or DelNS1-RBD with WT-RBD ($RBD_{WT}$, $n = 5$) or DelNS1 vector ($n = 5$). At weeks 4 and week 6, blood was collected from mice and tested for anti-S1 RBD-specific IgG titers by ELISA assay and neutralization activity against pseudovirus expressing wild-type SARS-CoV-2 spike protein. **F** Estimation of anti-RBD antibodies in mice prime-boost immunized intranasally with $2 \times 10^6$ pfu of DelNS1-RBD-DAF with Delta (B.1.617.2) RBD (Delta-DAF, $n = 8$) or Beta (B.1.351) RBD (Beta-DAF, $n = 8$), or DelNS1-RBD4N-DAF with Delta RBD4N (Delta4N-DAF, $n = 8$) or Beta RBD4N (Beta4N-DAF, $n = 8$), or DelNS1 vector ($n = 8$). At week 6, blood was collected from mice and tested for anti-S1 RBD-specific IgG titers and neutralization activity against pseudoviruses expressing spike proteins of SARS-CoV-2 variants. LOD: lower limit of detection. Error bars represent mean ± SD. Statistical analysis was performed using one-way ANOVA followed by Dunn's multiple comparisons test: ****$p < 0.0001$, **$p < 0.01$, *$p < 0.05$, ns not significant. Numerical labels indicate fold difference. Mouse cartoon created with BioRender.com.

Omicron RBD (Fig. 2B–D). To examine if DelNS1-RBD4N-DAF LAIVs induce mucosal immune responses, anti-RBD IgA was measured in bronchoalveolar lavage (BAL) fluid obtained from immunized mice. Our results showed that both Delta and Omicron RBDs expressed from DelNS1-RBD4N-DAF LAIVs induced IgA against the original strain of SARS-CoV-2 (Fig. 2E). In hamsters, Delta and Beta DelNS1-RBD4N-DAF intranasal vaccine candidates induced significantly higher levels of RBD-specific antibodies than did two doses of BioNTech mRNA vaccine (Fig. 2F). Similar to the results from immunized mice, neutralization assays using pseudovirus or live virus clearly showed strain-specific neutralization activity for each vaccine candidate in the hamster model (Fig. 2G, H). Consistent with other reports[26,27], parental SARS-CoV-2 (lineage A and basis for BioNTech mRNA vaccine), Delta and Beta spike epitopes induce few cross-neutralizing antibodies to the Omicron variants BA.1 and BA.2 using both pseudovirus and live virus (Fig. 2G, H). It is noted that BNT162b2 induced similar or higher levels of IgG in mice but lower levels of IgG in hamsters than Omi4N-DAF to the wild type RBD (Fig. 2B, F). It is speculated that cell from different species may vary in their mRNA vaccine intake and such difference may influence the immune response. These results indicate that RBDs from different variants of SARS-CoV-2 are immunogenic, inducing serum neutralizing antibodies and mucosal IgA targeting the RBD of SARS-CoV-2 spike proteins, with significant variation in strain specificity.

## DelNS1-RBD4N-DAF LAIVs induce CD4+ and CD8+ T cell responses

SARS-CoV-2 specific T cells play a critical role in clearing virus and preventing severe disease in humans[28–32]. Induction of RBD-specific T cells by DelNS1-RBD4N-DAF LAIVs representing Delta and Omicron variants of SARS-CoV-2 was evaluated in mice. Mice were immunized through the intranasal route with two doses of DelNS1 vector or DelNS1-RBD4N-DAF containing either the Delta or Omicron (BA.1) version of RBD, four weeks apart, and acute phase T cell responses in lungs and spleens estimated 10 days after boost vaccination (Fig. 3A). As expected, intranasal delivery triggers a greater T cell response in lungs than in spleens, reinforcing that specific airway immunity is better induced by intranasal immunization. While Delta DelNS1-RBD4N-DAF induces similar levels of CD4+ and CD8+ T cells, Omicron RBD (BA.1) appears to induce a strong CD4+ response but much lower levels of CD8+ T cells in both lungs and spleens of immunized mice (Fig. 3B, C). This may have been due to the peptide pool used, which was based on parental SARS-CoV-2 (lineage A strain). More variations from the parental (lineage A) strain were observed in the RBD region of Omicron than in Delta (Supplementary Table 2). It remains to be investigated if the Omicron variant induces different CD8+ T cell populations to the parental strain and previous variants of SARS-CoV-2. Decreased proliferative CD8+ but not CD4+ T cells recognizing Omicron compared with those specific for wild-type virus



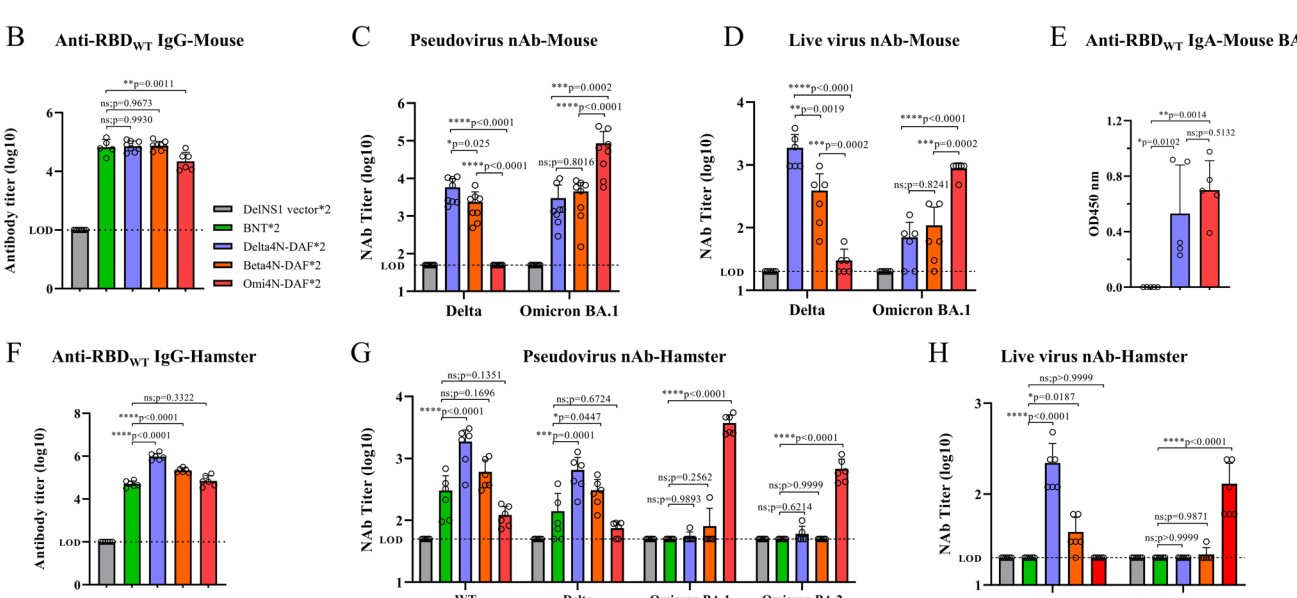

**Fig. 2 | Immunogenicity of DelNS1-RBD4N-DAF LAIVs in mice and hamsters.**
**A** Prime-boost immunization regimen and grouping of BALB/c mice and hamsters. BALB/c mice were prime-boost immunized intranasally with 2 × 10⁶ pfu of Delta4N-DAF, Beta4N-DAF, DelNS1-RBD4N-DAF with Omicron BA.1 RBD (Omi4N-DAF) or DelNS1 vector only control, or through intramuscular injections of the BioNTech BNT162b2 mRNA vaccine (BNT, 5ugand sera collected 14 days after the second immunization for testing of anti-S1 RBD-specific IgG titers (**B**) (Delta4N-DAF ($n = 8$ mice), Beta4N-DAF ($n = 8$ mice), Omi4N-DAF ($n = 6$ mice), DelNS1 vector ($n = 8$ mice), BNT ($n = 5$ mice)). **C** Neutralization titers against pseudotyped viruses displaying Delta or Omicron BA.1 spike proteins ($n = 8$ mice for each group), and (**D**) neutralization titers against live SARS-CoV-2 variants (Delta and Omicron BA.1) ($n = 6$ mice for each group). **E** The bronchoalveolar lavage (BAL) of mice was collected 10 days after boost immunization and anti-S1 RBD-specific IgA titers

determined ($n = 5$ mice for each group). Syrian hamsters were prime-boost immunized intranasally with either 5 × 10⁶ pfu of Delta4N-DAF, Beta4N-DAF, Omi4N-DAF, or DelNS1 vector only control, or through intramuscular injections of the BioNTech BNT162b2 mRNA vaccine (5ug). **F** Sera samples were collected 14 days after the second immunization and tested for anti-S1 RBD-specific IgG titers ($n = 6$ hamsters for each group). **G** Neutralization titers against pseudotyped viruses with wild type, Delta or Omicron spike proteins ($n = 6$ hamsters for each group), and (**H**) neutralization titers against live SARS-CoV-2 variants (Delta and Omicron BA.1) ($n = 6$ hamsters for each group). LOD lower limit of detection. Error bars represent mean ± SD. Numerical labels indicate means. Statistical analysis was performed using one-way ANOVA followed by Dunn's multiple comparisons test: ****$p < 0.0001$, ***$p < 0.001$, **$p < 0.01$, *$p < 0.05$, ns not significant.

have been observed among infected or vaccinated individuals[33]. To assess memory T cell responses, mice were prime-boost immunized with DelNS1 vector or Omi4N-DAF (BA.1 RBD) LAIVs and CD45-specific antibodies intravenously injected to label peripheral T cells immediately before mice were sacrificed. CD4⁺ and CD8⁺ memory phase T cells (CD45⁻) were detected in Omi4N-DAF immunized mice but not in vector-immunized controls (Fig. S1A, B). These results indicate that immunization with DelNS1-RBD4N-DAF through the intranasal route can induce acute and memory phase T cell responses in lungs and spleens, with much higher levels of acute phase T cells being present in the lungs.

**DelNS1-RBD4N-DAF LAIVs induce sterilizing immunity to SARS-CoV-2 infection**
After evaluating neutralizing antibodies and T cell responses, we examined protection against SARS-CoV-2 infection in mice and hamsters through immunization with DelNS1-RBD4N-DAF LAIVs. To facilitate challenge experiments using the mouse model, we made mouse-adapted (MA) SARS-CoV-2 strains, including Gamma (P.1) and Omicron BA.1 variants. These MA strains replicate efficiently and cause significant body weight loss and disease in mice (Fig. 4A, B and Fig. S2A, B). Immune evasion by Omicron variants has been extensively

reported in humans immunized with vaccines or through prior infection with earlier variants[5,26]. Delta-RBD induce lower levels of cross-variant neutralizing antibodies to Omicron BA.1 and BA.2 than Omicron-RBD (Fig. 2C, D, G). Notably, immunization with either Delta or Omicron (BA.1) DelNS1-RBD4N-DAF LAIVs fully prevented body weight loss in mice and protected them from SARS-CoV-2 infection in lung tissues, with virus being undetectable at days 2 and 4 following inoculation with a mouse adapted Omicron strain (BA.1) (Fig. 4). It is possible that other arms of immunity including specific T cells and mucosal IgA may act together to block and clear the challenged virus. In contrast, unimmunized mice lost more than 11% of their body weight and still had high viral titers in lungs on day 4 after virus challenge. A similar result was observed in mice immunized with Delta4N-DAF and challenged with mouse adapted SARS-CoV-2 Gamma (P1) variant (Fig. S2B, C). Hamster is one of the more susceptible laboratory animal models for SARS-CoV-2 infection, simulating the clinical and pathological manifestations of human COVID-19 disease[34,35]. We compared antibody induction in hamsters following prime-boost immunization with intramuscular prototype BNT162b2 mRNA or intranasal DelNS1-RBD4N-DAF (Omicron BA.1 or Delta RBD) vaccines, or prime-boost with BNT162b2 mRNA followed by a second boost with either BNT162b2 mRNA or a DelNS1-RBD4N-DAF intranasal vaccine (Fig. 5A

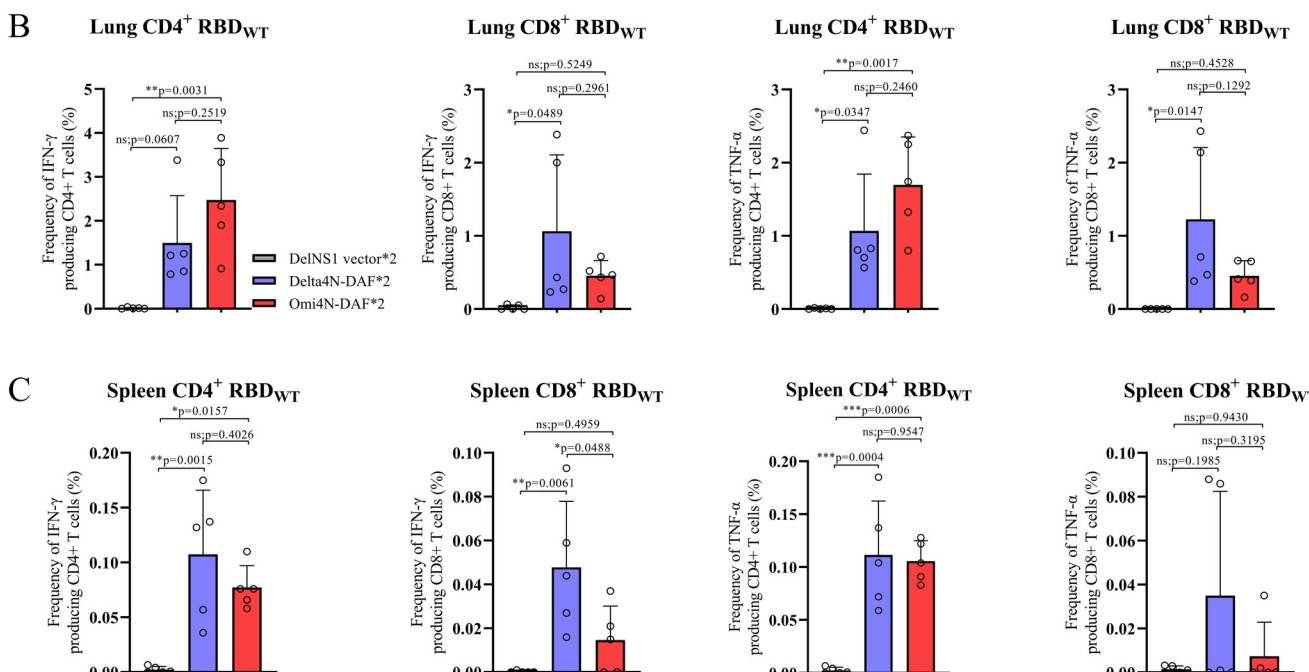

**Fig. 3 | DelNS1-RBD4N-DAF LAIV induces CD4+ and CD8+ T cell response.**
**A** Schedule for immunization of BALB/c mice. At day 10 after the second immunization, lung cells and splenocytes were obtained and stimulated with or without spike peptide pools (Supplementary Table 2) overnight in the presence of BFA. Surface markers (CD4, CD8, and Zombie) were stained, and cells were fixed and permeabilized. Intracellular cytokines (TNFα and IFNγ) were then stained with specific antibodies. Delta4N-DAF and Omi4N-DAF induced IFN-γ and TNF-α positive CD4+ and CD8+ T cell responses in lungs (**B**) and spleens (**C**) of BALB/c mice. Percentages of IFNγ+ or TNFα+ CD4+ and CD8+ T cells in immunized mice (*n* = 5 for each group) were compared. Error bars represent mean ± SD. Statistical analysis was performed using one-way ANOVA followed by Dunn's multiple comparisons test: \*\*\**p* < 0.001, \*\**p* < 0.01, \**p* < 0.05, ns not significant. Mouse cartoons created with BioRender.com.

and Fig. S3A). All vaccination schemes induced fairly comparable levels of antibodies against the RBD of the parental SARS-CoV-2 strain (lineage A) (Fig. S3B). Pseudovirus-based neutralization assays provided information for understanding specific neutralizing activity induced by each immunization combination. Consistent with an earlier report[36], the Delta VOC largely evades antibodies induced by two doses of BNT162b2 mRNA while a 3rd dose booster of either intramuscular BNT162b2 mRNA or intranasal DelNS1-RBD4N-DAF vaccine (Delta or Omicron BA.1 RBD) induces neutralizing activity against the Delta VOC (Fig. S3C), as does a double dose of Delta4N-DAF (Fig. 2G). However, two doses of Omi4N-DAF intranasal vaccine induced only low levels of neutralizing antibodies against the Delta variant (Fig. 2G). In contrast, all combinations except two doses of Omi4N-DAF vaccine failed to elicit significant levels of neutralizing antibodies against the Omicron variants (BA.1 & BA.2) in hamsters (Figs. 2G, H and 3C).

We next evaluated protection by intranasal immunization with DelNS1 vector, Delta4N-DAF or Omi4N-DAF LAIVs and compared these to intramuscular immunization with BioNTech mRNA vaccine in SARS-CoV-2 challenged hamsters. Delta, Omicron BA.1 and Omicron BA.2 variants were used in challenge experiments to understand the cross protection afforded by the tested DelNS1-RBD4N-DAF vaccines (Fig. 5A). Challenge with Delta variant caused significant body weight loss (about 6%) in control and most groups of immunized hamsters except for those immunized with Delta4N-DAF (Fig. 5B and Fig. S3D), which completely prevented body weight loss. Consistent with the preservation of body weight, immunization with Delta4N-DAF LAIV prevented virus infection

in the lungs and nasal turbinates of Delta-challenged hamsters (Fig. 5C). Intramuscular prime-boost immunization with BNT162b2 mRNA vaccine (based on lineage A strain) or BNT162b2 mRNA prime-boost followed by a second boost with either intramuscular BNT162b2 or intranasal Delta4N-DAF vaccine reduced Delta variant replication 10-fold or more in the lungs and nasal turbinates (Fig. S3E). However, immunization with Omi4N-DAF LAIV failed to block Delta replication in hamsters (Fig. 5C). We then tested whether Omi4N-DAF LAIV could prevent infection with Omicron variants. Omicron variants cause milder disease in humans and are less pathogenic in animal models[6,37]. When immunized hamsters were challenged with Omicron BA.1 slight body weight loss (2.65%) was observed in control, BNT162b2 mRNA, prime-boost BNT162b2 mRNA combined with a second booster of Omi4N-DAF intranasal vaccine and Omi4N-DAF immunized hamsters, while no apparent body weight loss was observed in the Delta4N-DAF immunized group (Fig. 5D and Fig. S3F). Virus titration in lung tissues and nasal turbinates revealed that only vaccination with strain-specific Omi4N-DAF completely prevented virus replication in the lungs, with only very low levels of virus being detected in the nasal turbinates of one of the Omi4N-DAF-immunized hamsters (Fig. 5E). Interestingly, immunization with Delta4N-DAF also significantly reduced virus replication in the lungs of Omicron-infected animals and completely blocked virus replication in the nasal turbinates (Fig. 5E), suggesting a significant cross protective effect against the Omicron variant being conferred by nasal immunization with Delta4N-DAF. However, immunization with two or three doses of intramuscularly injected BNT162b2 mRNA vaccine caused no apparent reduction in virus

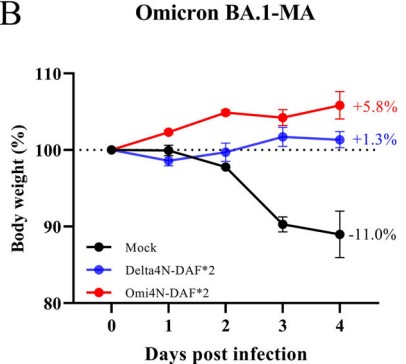
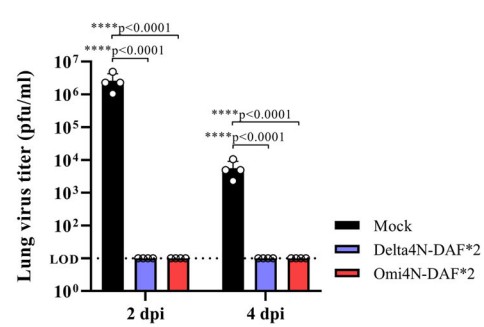

**Fig. 4 | DelNS1-RBD4N-DAF LAIVs protect mice challenged with mouse-adapted Omicron SARS-CoV-2 variant. A** Illustration of schedule of immunization, SARS-CoV-2 virus challenge and sacrifice for BALB/c mice. 6–8-week-old BALB/c mice were intranasally prime-boost vaccinated with Delta4N-DAF ($2 \times 10^6$ pfu) or Omi4N-DAF ($2 \times 10^6$ pfu) or PBS ($n = 8$ for each group) and then challenged with a mouse-adapted Omicron BA.1 SARS-CoV-2 strain (Omicron-MA, $1 \times 10^5$ pfu) 4 weeks after boost immunization. **B** Body weight changes following virus challenge ($n = 8$ for each group). **C** Virus titers in the lungs were measured at 2 dpi ($n = 4$ for each group) and 4 dpi ($n = 4$ for each group). MA: mouse-adapted. LOD: lower limit of detection. Error bars represent mean ± SD. Statistical analysis was performed using one-way ANOVA followed by Dunn's multiple comparisons test: ****$p < 0.0001$. Mouse cartoon created with BioRender.com.

titers in the lungs and nasal turbinates of Omicron BA.1-challenged hamsters, while two doses of BNT162b2 plus one dose of Omi4N-DAF intranasal booster reduced viral titers in the lungs 100-fold (Fig. S3G). The Omicron BA.2 sub-lineage and its descendants (BA.2.12.1 and BA.4/5) are more transmissible than Omicron BA.1 and have become the dominant strain in many countries since early 2022[38]. We compared the effectiveness of BNT162b2 mRNA and Omi4N-DAF vaccination in hamsters subsequently challenged with the Omicron BA.2 strain (Fig. 2A and Fig. S4A). Challenge with Omicron BA.2 led to more than 6% body weight loss in mock-immunized hamsters, while immunization with BNT162b2 mRNA and Omi4N-DAF reduced body weight loss to a certain extent, with a stronger effect being observed with the Omi4N-DAF, three-dose BNT162b2 mRNA and two-dose BNT162b2 plus Omi4N-DAF booster vaccination schedules (Fig. 5F and Fig. S3H). Determination of virus titers in respiratory tissues revealed that only Omi4N-DAF vaccination completely blocked virus replication, with lungs and nasal turbinates being virus-free at day 4 post virus challenge, while other vaccination combinations failed to inhibit virus replication in the lungs and did not block virus shedding in the nasal turbinates (Fig. 5G), while BNT162b2 mRNA (two and three doses) or two doses of BNT162b2 mRNA plus one Omi4N-DAF intranasal booster significantly reduced virus titers in the lungs of infected animals but not in the nasal turbinates (Fig. 5G & Fig. S3I). Histopathological examination confirmed the protective effects of the different vaccines in mice and hamsters (Fig. S5A–C and S6A–D). These results demonstrate that nasal immunization with influenza virus-vectored DelNS1-RBD4N-DAF vaccines can provide near-sterilizing immunity to SARS-CoV-2 infection. Cross-variant immunity induced by earlier variants against the most recent Omicron VOC is also possible with this vaccine system.

### DelNS1-RBD4N-DAF LAIV can be used as a dual function vaccine for influenza and SARS-CoV-2

SARS-CoV-2 cannot be eradicated, and it is likely to continue to evolve and co-circulate with other coronaviruses and influenza viruses in humans. It is possible that co-circulating SARS-CoV-2 and influenza viruses may cause annual epidemics. Regular vaccination to boost immunity to emerging SARS-CoV-2 variants may be necessary, as it is for seasonal influenza. A dual function vaccine for both influenza and SARS-CoV-2 may be an option in the future management of these viruses. We have previously shown that DelNS1 live attenuated virus has potential to be developed as another kind of live attenuated influenza vaccine, due to its highly attenuated and immunogenic properties[16]. This study further tested if DelNS1-RBD4N-DAF LAIVs, which we showed above to protect against SARS-CoV-2, can also be used for protection against influenza virus infection. CA4-DelNS1-RBD (H1N1) and HK68-DelNS1-RBD (H3N2) LAIVs were used to immunize mice before challenge with mouse adapted H1N1 or H3N2 viruses, respectively. Our result showed that prime-boost immunization with CA4-DelNS1-RBD (H1N1) and HK68-DelNS1-RBD (H3N2), respectively, fully protected mice challenged with a lethal dose of H1N1 or H3N2 mouse adapted virus, and completely blocked virus replication in lung tissues (Fig. 6A–E). Examination of specific antibodies to hemagglutinin (HA) and T cells recognizing the nucleoprotein (NP) of influenza virus showed that DelNS1-RBD4N-DAF LAIVs retain the ability to induce immunity to influenza components in immunized mice (Fig. 6F, G). To assess the potential influence or interference of pre-existing anti-influenza immunity on immunogenicity and protective ability of the Omicron DelNS1-RBD4N-DAF LAIV, we tested the effect of prior influenza virus infection on our vaccine and found the prime-boost immunization scheme used is able to induce robust SARS-CoV-2 specific antibodies and neutralizing antibodies in mice previously infected with sublethal doses of influenza strains matching the DelNS1-RBD4N-DAF vaccines (Fig. S4A–E). Mice were fully protected after challenge with a mouse-adapted Omicron BA.1-MA strain of SARS-COV-2, no detectable virus titer in the lung tissues of mice on day 2 and 4 post infection (Fig. S4F, G). These results demonstrate that the DelNS1-RBD4N-DAF influenza virus vector vaccine platform has strong potential to be developed into a dual function vaccine for prevention of infection with both influenza and SARS-CoV-2 viruses.

### Discussion

SARS-CoV-2 continues to circulate in humans through the emergence of variants with immune evasion ability and optimal host adaptation

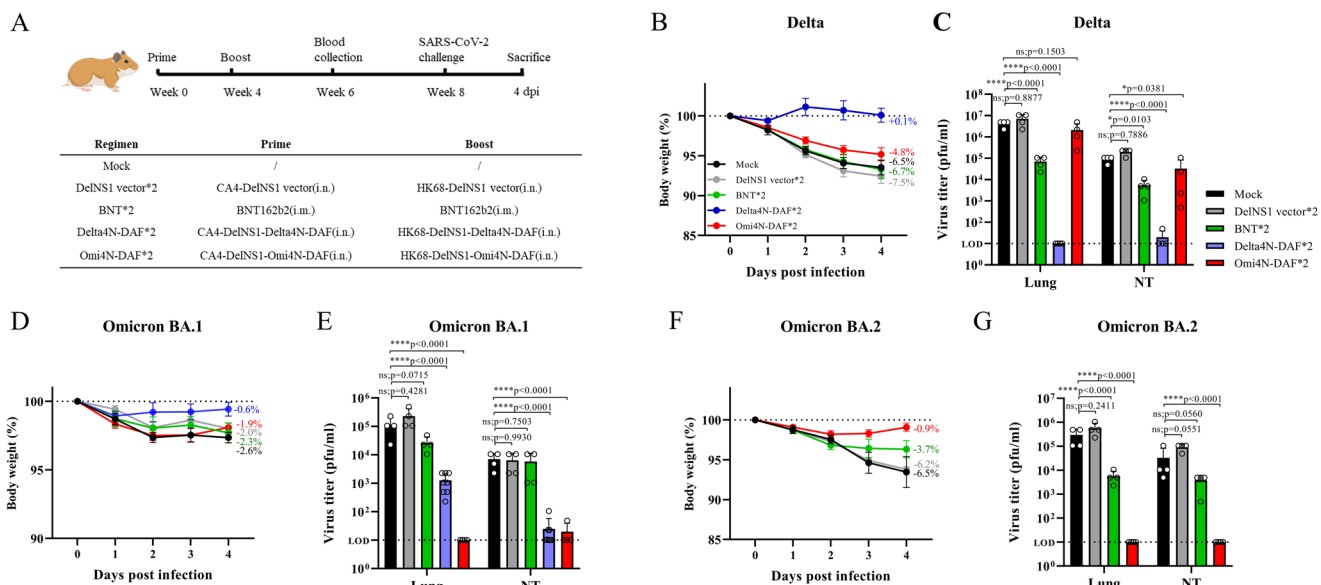

**Fig. 5 | Protection against SARS-CoV-2 virus challenge in hamsters through prime-boost immunization with BNT162b2 mRNA or DelNS1-RBD4N-DAF vaccines. A** Illustration of schedule of immunization, blood collection and SARS-CoV-2 virus challenge for hamsters. Hamsters were prime-boost vaccinated either intranasally with Delta4N-DAF ($5 \times 10^6$ pfu), Omi4N-DAF ($5 \times 10^6$ pfu), DelNS1 vector ($5 \times 10^6$ pfu) or PBS (mock) or intramuscularly with BNT162b2 mRNA (1/6 clinical dose (5ug)) vaccine. Hamsters were challenged with SARS-CoV-2 variants Delta or Omicron BA.2 at $1 \times 10^4$ pfu per hamster, 4 weeks after boost immunization. Body weight changes following SARS-CoV-2 virus challenge of hamsters immunized with Delta4N-DAF, Omi4N-DAF or BNT162b2 mRNA vaccines or controls (n = 4 for each group) (**B**, **D**, and **F**). Virus titers in the lungs and nasal turbinates (NT) of hamsters were measured at 4 dpi (n = 4 for each group) (**C**, **E**, and **G**). NT nasal turbinates. LOD lower limit of detection. Error bars represent mean ± SD. Statistical analysis was performed using one-way ANOVA followed by Dunn's multiple comparisons test: ****$p < 0.0001$, *$p < 0.05$, ns not significant. Hamster cartoon created with BioRender.com.

for more efficient transmission[39]. The global population is gradually building up basic immunity to SARS-CoV-2 infection through either vaccination or acquired infection. However, active circulation of SARS-CoV-2 variants still leads to substantial numbers of reinfections[5], impacting human health. Booster vaccinations with current vaccines have helped to alleviate COVID-19 disease but will not provide long term immunity to keep the prevalence of emerging variants of SARS-CoV-2 at low levels[40]. The next generation of COVID-19 vaccines should focus on preventing infection and/or reducing transmission. Intranasally delivered vaccines that induce stronger mucosal immunity are one such direction for vaccine development[41,42]. One recent study showed no association between nasal IgA and plasma IgG anti-S1 (Spike) responses and suggested that intranasal boost vaccination is necessary to enhance nasal immunity and reduce reinfection[42]. Although clinical trials of some intranasal vaccine candidates are currently being conducted[20], an early version of our influenza-based DelNS1-RBD LAIV intranasal vaccine for COVID-19 has recently been approved for emergency use in humans in China. We previously reported influenza-based DelNS1-RBD vaccines, demonstrating that prime-boost immunization with these intranasal vaccines or their use together with an intramuscular vaccine candidate induced strong systemic and mucosal immune responses that blocked SARS-CoV-2 infection in a mouse model[14,18]. We have further evaluated a modified version of the influenza-based viral vector, DelNS1-RBD4N-DAF, in mouse and hamster models in this study. The current version carries two important features; the addition of a membrane anchoring motif, DAF, that optimizes cell surface RBD expression and introduction of four N-glycosylation sites to shield epitopes outside of the receptor-binding motif (RBM) and encourage generation of antibodies specific for ACE2 competing epitopes (Fig. 1). We showed that DelNS1-RBD4N-DAF LAIVs expressing various RBDs derived from variants of SARS-CoV-2 are immunogenic and able to induce both systemic and mucosal specific immunity against SARS-CoV-2 RBD and effectively protect, and in some instances cross-protect, against infection by SARS-CoV-2

variants in mouse and hamster models (Figs. 1–5 and Fig. S1–S6). These influenza-based DelNS1-RBD4N-DAF vaccines distinguish themselves through their ability to induce immunity in respiratory tissues and to provide near-sterilizing immunity against SARS-CoV-2 infection.

Intranasal immunization specifically induces mucosal immunity in the upper respiratory tissues[13,43–45]. One recent study reported an intranasal Ad-vectored vaccine expressing spike, nucleocapsid, and RdRp antigens that induced complex immunity against SARS-CoV-2 challenge superior to that of intramuscular immunization in a mouse model[46]. The approval of the first intranasal spray vaccines against COVID-19 in China in December 2022 marked a significant milestone in the development of this technology. About two decades ago, a live attenuated influenza vaccine based on cold adaptation mutants was first used in humans[47]. The influenza virus has not previously been used as a vaccine vector for other viral diseases, partly due to its compact and segmented viral genome. Identification of adaptative mutations in NS1-deleted (DelNS1) influenza viruses that allow replication in embryonated chicken eggs and MDCK cells provided the opportunity to use influenza virus for the development of new live attenuated vaccines for influenza and other viral respiratory infections[16]. Because the NS1 protein of the influenza virus is a key virulence element and immune modulator[48,49], it follows that removal of the NS1 gene from the influenza virus will make the NS1-deleted mutant virus a better immunogen for triggering the host immune response. Indeed, previous studies of ours and others showed that deletion of NS1 from the viral genome enhances the immunogenicity of DelNS1 live attenuated virus in animal models[16,50,51]. The inclusion of DAF for cell surface presentation and selective glycosylation of non-ACE2 competing epitopes in the new DelNS1-RBD4N-DAF LAIV vector described in this study have significantly enhanced immunogenicity (Fig. 1). Intranasal immunization with DelNS1-RBD4N-DAF LAIV mimics natural influenza virus infection to deliver the RBD antigen to upper respiratory tissues. Deletion of NS1 makes this influenza viral vector extremely safe for human use, as shown in phase I/II clinical trials[19]. Notably, the DelNS1

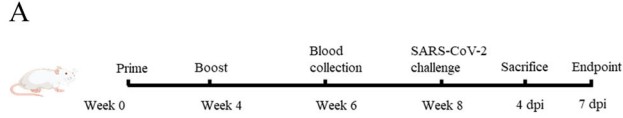

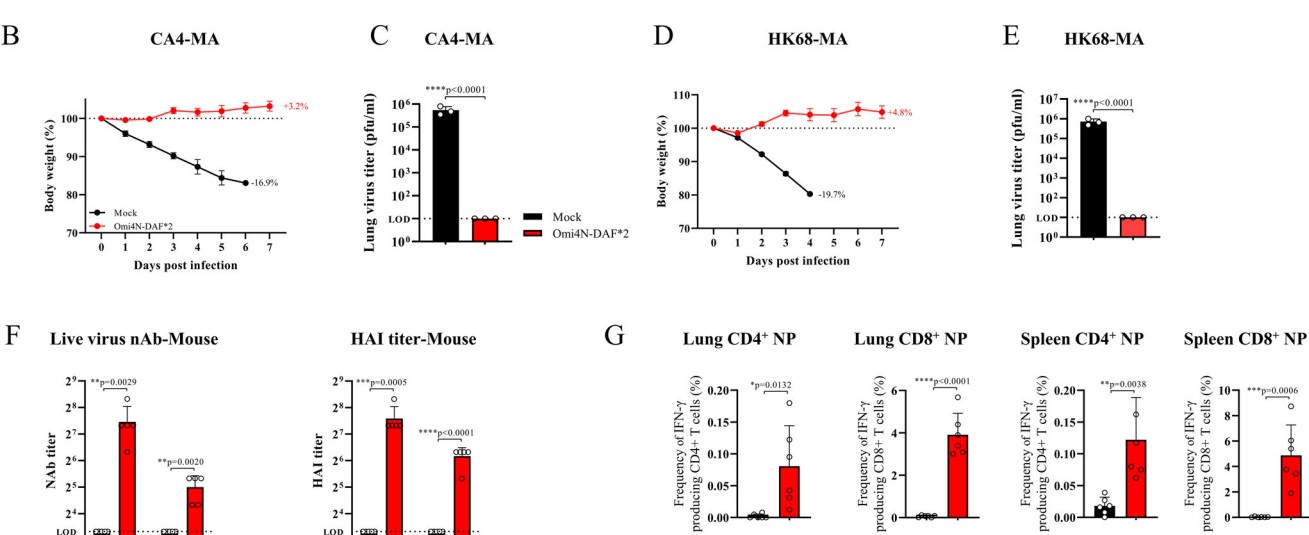

**Fig. 6 | Protection against influenza virus challenge in mice prime-boost immunized with DelNS1-RBD4N-DAF LAIV. A** Illustration of schedule of immunization, blood collection and influenza virus challenge for BALB/c mice. BALB/c mice were intranasally prime-boost vaccinated with Omi4N-DAF ($2 \times 10^6$ pfu) or PBS ($n = 6$ for each group), and then challenged with mouse-adapted influenza virus strains matching the influenza subtype of vaccines (CA4-) (H1N1, $5 \times 10^3$ pfu) or HK68-MA (H3N2, $1 \times 10^4$ pfu) ($n = 3$ for each group)) 4 weeks after boost immunization. Body weight changes in mice following influenza virus challenge were tracked ($n = 6$ for each group) until the control group reached the body weight loss cut-off (20%) for euthanization, in accordance with animal ethics protocols (**B**, **D**). Virus titers in the lungs were measured at 4 dpi (**C**, **E**). **F** BALB/c mice were prime-boost immunized intranasally with $2 \times 10^6$ pfu of Omi4N-DAF or PBS (mock) ($n = 5$ for each group). Sera were collected 14 days after the second immunization for testing of neutralization titers and hemagglutination inhibition (HAI) titers against

live influenza viruses CA4 (H1N1) or HK68 (H3N2). **G** At 5 weeks after the second immunization, 2 μg of PerCP-Cy5.5 conjugated CD45-specific antibody was injected i.v. via the tail vein 5 min before sacrifice. Lung cells and splenocytes were obtained and stimulated with or without influenza NP peptides, overnight in the presence of BFA. Surface markers (CD69, CD103, CD4, CD8, and Zombie) were stained, and cells then fixed and permeabilized. Intracellular IFNγ was then stained with specific antibodies. Omi4N-DAF induced NP-specific tissue-resident memory T (Trm) cell responses in lungs (CD45- IFN-γ+ CD69+ CD4+ T cells and CD45- IFN-γ+ CD69+ CD103+ CD8+ T cells) and spleens (CD45- IFN-γ+ CD4+ and CD8+ T cells). Percentages of T cell subsets in immunized ($n = 6$) and mock ($n = 6$) groups were compared. MA mouse-adapted. HAI hemagglutination inhibition. LOD lower limit of detection. Error bars represent mean ± SD. Statistical comparisons between means were performed by Student's $t$ test (2-tailed): $****p < 0.0001$, $***p < 0.001$, $**p < 0.01$, $*p < 0.05$. Mouse cartoons created with BioRender.com.

LAIV was found to induce high levels of interferon β in infected cells[17]. Therefore, this vector can also serve as an adjuvant to enhance the immune response against antigens (in this case, RBD of SARS-CoV-2 expressed from the NS1 site of the influenza virus genome). One of the important properties of this system, compared to currently used vaccines for COVID-19, is the ability to induce immunity that prevents disease and blocks virus replication in the upper respiratory tract, an attribute urgently needed in the next generation of vaccines to slow down or stop transmission of SARS-CoV-2 variants. Most importantly, by incorporating HA and NA matching circulating influenza strains, DelNS1-RBD4N-DAF LAIVs can be constructed to make bi-functional vaccines in response to seasonal influenza and SARS-CoV-2 dual epidemics in the future.

## Methods

### Construction and generation of DelNS1-RBD-DAF LAIV vaccines

CA4-DelNS1 is the backbone viral vector, derived from A/California/4/2009 (H1N1), from which the NS1 has been deleted. HK68-DelNS1 was made from the CA4-DelNS1 backbone but with HA and NA being derived from A/Hong Kong/1/1968 (H3N2)[16]. In the NS segment, the NS1 gene was deleted and replaced with SARS-CoV-2 spike RBD sequence. The signal peptide sequence of tissue plasminogen activator (tPA) was added to the N terminal of RBD and the sequence for the

transmembrane domain of decay accelerating factor (DAF) was added to the C terminal. The DAF peptide (33 amino acids) includes the transmembrane domain and cytoplasmic tail of the DAF protein. The addition of the signal peptide and the DAF transmembrane sequences to the RBD ensure its proper processing and expression on the cell surface. To direct the immune response to the most important receptor-binding motif (RBM), 4 glycosylation sites were introduced into the RBD to block the non-neutralizing epitopes outside the RBM. Four mutations (A372T, G413N, D428N, P521N) were introduced using a QuikChange kit (Agilent). For virus rescue, eight pHW2000 plasmids containing a version of the NS segment (DelNS1, DelNS1-RBD, DelNS1-RBD-DAF or DelNS1-RBD4N-DAF with RBD derived from WT (lineage A virus), Beta (B.1.351), Delta (B.1.617.2) or Omicron BA.1 strains), HA and NA surface protein segments from A/California/4/2009 (H1N1) or A/Hong Kong/1/1968 (H3N2) and the other 5 influenza virus genomic segments from the CA4 backbone together with an NS1 expression plasmid, were transfected into HEK293T cells using TransIT-LT1 (Mirus) according to the manufacturer's protocol. After overnight incubation at 33 °C, the DNA mix was removed and MEM supplemented with 1 μg/ml TPCK-treated trypsin (Sigma) was added. Virus supernatant was collected 72 h later and designated passage 0 (P0) virus. Virus was passaged in embryonated chicken eggs for 48 h at 33 °C and then aliquoted and titrated by plaque assay using MDCK

cells. In this way, two groups of different subtype LAIVs, CA4-DelNS1-RBD (H1N1) and HK68-DelNS1-RBD (H3N2) were generated, together with empty vector control viruses without NS1 or RBD (CA4-DelNS1 and HK68-DelNS1).

To assess the effect of N-glycosylation on RBD expression in DelNS1-RBD4N-DAF LAIV, MDCK (ATCC-CCL-34), BHK21(ATCC-CCL-10) and A549 (ATCC-CCL-185) cells were mock infected or infected with DelNS1-RBD (WT, lineage A virus), DelNS1-RBD with individual N-glycosylation mutations (A372T, G413N, D428N, or P521N), or DelNS1-RBD4N (4N) LAIV virus at an MOI of 1 for 10 h. Cell lysates were analyzed by western blot using anti-RBD antibody and anti-NP antibody.

Animal immunization and SARS-CoV-2 challenge.

For mouse immunization, 6–8-week-old female BALB/c mice were anesthetized with ketamine and xylazine and then immunized intranasally with $2 \times 10^6$ pfu of either CA4DelNS1-RBD (RBD (WT, lineage A virus), RBD-DAF (WT, Delta or Beta) or RBD4N-DAF (Delta, Beta or Omicron)) or empty vector (CA4-DelNS1), or with PBS. After 4 weeks, mice were then boosted with HK68-DelNS1-RBD (RBD (WT), RBD-DAF (WT, Delta or Beta) or RBD4N-DAF (Delta, Beta or Omicron)) or controls (empty vector (HK68-DelNS1) or PBS). For comparison in one experiment, Pfizer-BioNTech (called BioNTech or BNT162b2 henceforth) mRNA vaccine was used to immunize mice intramuscularly at one sixth of the standard human dose (5 µg per mouse). Blood sera was collected at selected time-points for ELISA and neutralization assays. Ten days after boost immunization, some mice were euthanized and bronchoalveolar lavage fluid, lungs and spleens collected for ELISA and T cell assay. Four weeks after boost immunization, the remaining mice were challenged intranasally with mouse adapted SARS-CoV-2 strains (Gamma-MA: $5 \times 10^4$ pfu, Omicron-MA: $1 \times 10^5$ pfu) or mouse adapted influenza strains (CA4-MA: $5 \times 10^3$ pfu, HK68-MA: $1 \times 10^4$ pfu). Body weight and disease symptoms were monitored daily. At days 2 and 4, lungs were collected for virus titration and histopathological study.

To assess the effect of pre-existing anti-influenza immunity, six-to eight-week-old female BALB/c mice were challenged with a sublethal dose containing both wild type CA4 and HK68 viruses ($2 \times 10^4$ pfu of each virus per mouse) or control PBS. Body weight and disease symptoms were monitored for 2 weeks. At 4 weeks post infection, sera were collected to determine antibody responses against the two WT (lineage A virus) influenza viruses. Mice were then primed with CA4-DelNS1-Omi4N-DAF ($2 \times 10^6$ pfu per mouse). After 4 weeks, mice were boosted with HK68-DelNS1-Omi4N-DAF ($2 \times 10^6$ pfu per mouse). As an additional control, a group of control (PBS) mice were sequentially immunized with the empty viral vectors CA4-DelNS1 and HK68-DelNS1. Sera were again collected to determine antibody responses against SARS-CoV-2 Omicron virus. After serum collection, mice were challenged with Omicron-MA virus ($1 \times 10^5$ pfu per mouse). Body weight and disease symptoms were monitored for 4 days. At day 2 and day 4 post infection, lungs were collected for viral titer determination.

For hamster immunization, 6–8-week-old male golden Syrian hamsters were anesthetized with ketamine and xylazine and then immunized intranasally with $5 \times 10^6$ pfu of either CA4-DelNS1-RBD (RBD4N-DAF (Delta, Beta or Omicron)) or CA4-DelNS1 empty vector, or with PBS control. For comparison, Pfizer-BioNTech mRNA vaccine was used to immunize hamsters intramuscularly at one sixth of the standard human dose (5 µg per hamster). After 4 weeks, hamsters were boosted with HK68-DelNS1-RBD vaccines, the BioNTech vaccine or controls, as appropriate. After a further 4 weeks some groups received a second booster immunization. Blood sera was collected at selected time-points for ELISA and neutralization assays. Four weeks after boost immunization, hamsters were challenged intranasally with different SARS-CoV-2 variants at $1 \times 10^4$ pfu per hamster. Body weight and disease symptoms were monitored daily. At day 4, lungs

and nasal turbinates were collected for virus titration and histopathological study.

All animal experiments were approved by the Committee on the Use of Live Animals in Teaching and Research of the University of Hong Kong (HKU) (CULATR) (# 5512-20, 5359-20, 5377-20). All animal experiments related to SARS-CoV-2 were performed in a biosafety level 3 laboratory at HKU.

### Generation of mouse adapted SARS-CoV-2 viruses
To facilitate experiment using mouse model, we made mouse adapted SARS-CoV-2 from Gamma (P1) and Omicron (BA.1) variants which contain N501Y mutation in the RBD region of spike and can be easily adapted to infect mice. Female BALB/c mice were infected with either Gamma or Omicron variant SARS-CoV-2 viruses at $1 \times 10^5$ pfu per mouse. After 2 days, lungs were collected and homogenized in 1 ml PBS, and 50 µl of lung homogenate then inoculated into a naive mouse. Passaging was repeated until the indicated passage number. For each passage, lung virus titers were determined by $TCID_{50}$ assay. Viruses obtained from the last passage were sequenced using the Sanger method. The sequence of mouse adapted SARS-CoV-2 viruses have been submitted to the GenBank (accession numbers: OQ619133 and OQ619134) and GISAID Database under accession numbers EPI_ISL_12996408 (Omicron-MA) and EPI_ISL_12996407 (Gamma-MA) respectively. Mutations identified in the mouse-adapted strains are included in Supplementary Table 1.

### Enzyme-linked immunosorbent assay (ELISA)
Anti-spike WT (lineage A virus) RBD IgG and IgA detection kits were gifts from Wantai Company. Procedures were conducted according to the manual. Briefly, heat inactivated sera or bronchoalveolar lavage fluid were 10-fold serially diluted and added to the plate and incubated at 37 °C for 30 min. The plate was washed 5 times and then incubated with secondary antibody reagent at 37 °C for 30 min. After washing, color development solution was added and incubated at 37 °C for 15 min. Stop solution was added and absorbance at 450 nm was measured using a Victor3 plate reader (PerkinElmer).

### Pseudovirus neutralization assay
Blood sera were heat inactivated at 56 °C for 30 min prior to use. The plasmids encoding different spike variants were constructed using vesicular stomatitis virus (VSV). To generate SARS-CoV-2 spike pseudotyped virus, HEK293T (ATCC-CRL-3216) cells were transfected with different spike expression plasmids and pNL4-3Luc_Env_Vpr (human immunodeficiency virus type 1 backbone). After 48 h, viral supernatant was aliquoted and stored at −80 °C. Blood sera were serially diluted and incubated with the appropriate amount of pseudovirus at 37 °C for 1 h. The mixtures were then added into HEK293T-hACE2 cell cultures. Two days later, cells were lysed and luciferase activity was measured using Luciferase Assay System kits (Promega) with a Victor3 plate reader. The 50% inhibitory concentration ($IC_{50}$) was calculated using GraphPad Prism.

### SARS-CoV-2 live virus neutralization assay
Heat inactivated sera were 2-fold serially diluted in DMEM medium and incubated with 100 pfu of the indicated virus at 37 °C for 1 h. The mix was added to confluent Vero-TMPRSS2 cells and incubated for 4 days at 37 °C. After 4 days, cytopathic effects (CPE) were assessed under a microscope, with the neutralization endpoint being the highest dilution with 50% inhibition of CPE.

### Intracellular cytokine staining (ICS)
Lungs and spleens of mice were collected following immunization. Splenocytes were isolated and homogenized through cell strainers (BD) and resuspended in RPMI medium (10% FBS and P/S). Lung tissue was chopped and digested in RPMI solution with collagenase

II (1 mg/ml) (Sigma) for 1 h at 37 °C. Red blood cells were lysed by addition of Lysing solution (BD). After washing with RPMI, cells were counted and resuspended in RPMI. Cells were stimulated with RPMI solution containing 1 mg/ml of either WT (lineage A virus) RBD peptide pool (15-mers overlapping by 11 residues to cover the whole RBD) or influenza A virus NP peptides or no peptide control (Supplementary Table 2)[16]. After 1 h, brefeldin A (BFA) was added to cells and incubated overnight. Cells were then washed with FACS buffer (2% FBS in PBS) and stained with anti-CD8-PerCP-Cy5 (Biolegend-100734), anti-CD4-V450 (BD-560468) and Zombie (BioLegend) for 30 min at 4 °C. Cells were washed and fixed with Perm/wash buffer (BD), and then stained with anti-IFNγ-APC (Biolegend-505810) and anti-TNFα-PE (BioLegend-506306) overnight at 4 °C. Cells were then washed twice with FACS buffer and resuspended in FACS buffer. The samples were acquired with a BD FACSAria III cell sorter and the generated data analyzed with FlowJo V9. Cytokine production was calculated by subtracting the no peptide control background value.

To study tissue resident memory T cells, 5 weeks after the boost immunization, 2 µg of anti-CD45-PerCP-Cy5.5 (BD-550994) was injected i.v. 5 mins before sacrifice. Lung and spleen cells were processed and stimulated as above. For surface marker staining, anti-CD69-BV711 (Biolegend-104537) and CD103-BV421 (Biolegend-121422) were included in addition to staining for the original markers (Zombie, anti-CD4-FITC (Biolegend-100406), anti-CD8-APC-Fire750 (Biolegend-100766)). Tissue resident (non-circulating) cells are CD45-. The gating strategy is shown in Fig. S7.

### Plaque assay and TCID$_{50}$ assay
For plaque assay, confluent Vero-TMPRSS2 cells in 6 well plates were incubated with 10-fold serially diluted virus for 1 h. The virus was then discarded, and cells washed and overlaid with 1% agarose DMEM and incubated for 3 days at 37 °C. Cells were fixed with 10% formaldehyde for 1 day. Agarose gels were removed, and plaques stained with 1% crystal violet and counted.

For TCID$_{50}$ assay, confluent Vero-TMPRSS2 cells in 96 well plates were incubated with serially diluted virus. After 4 days, cytopathic effects (CPE) were detected under a microscope. TCID$_{50}$ values were calculated by the Reed–Muench Method.

### Histopathology and immunofluorescence (IF) staining
Organs were fixed in 10% PBS buffered formalin and processed into paraffin-embedded blocks. Tissue sections were stained with haematoxylin and eosin (H&E) and examined by light microscopy. For IF staining, MDCK cells were mock infected or infected with DelNS1, DelNS1-RBD, or DelNS1-RBD-DAF viruses at an MOI of 1 for 10 h and cells processed using antibodies specific for RBD (red). Nuclei were stained with DAPI (blue).

### Statistical analysis
Statistical analysis was carried out using GraphPad Prism. Data were presented as the mean values ± SD of at least 3 replicates, unless otherwise indicated. Statistical significance was analyzed by either Student's $t$ test or one way analysis of variance (ANOVA) followed by Dunn's multiple comparisons test. For all tests: ****$p < 0.0001$, ***$p < 0.001$, **$p < 0.01$, *$p < 0.05$, ns−not significant.

### Reporting summary
Further information on research design is available in the Nature Portfolio Reporting Summary linked to this article.

### Data availability
All data associated with this study are presented within the paper or in the Supplementary materials. A sequence of mouse adapted SARS-CoV-2 strains are available in GenBank under accession numbers OQ619133 and OQ619134). Raw data underling the results are provided with this paper. Source data are provided with this paper.

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

## Acknowledgements

The authors would like to thank Dr. Jane Rayner for critical reading and editing of the manuscript. This study is partly supported by the Theme-Based Research Scheme (T11-709/21-N), the Collaborative Research Fund (C5110-20GF), and the General Research Fund (17107019) of the Research Grants Council, and HMRF Commissioned Research on COVID-19 (COVID1903010, COVID190123, and HMRF19181052), Hong Kong Special Administrative Region, China, and the Emergency Collaborative Project (EKPG22-01) of Guangzhou Laboratory.

## Author contributions

P.W. and H.C. conceived the studies; P.W., S.D., Y.L., R.C-Y. T., P.C., B.W-Y.M., T.L., A.J.Z., R. Z., H.X., A.K., and W.S. performed experiments; P.W., S.D., Y.L., A.J.Z., K.K-W.T., J.F-W.C., Z. C., K-Y.Y., and H.C. analyzed and interpreted the data; P.W., S.D., and H.C. wrote the paper.

## Competing interests

The authors declare that the University of Hong Kong has filed patents on work related to the generation and application of DelNS1 live attenuated influenza vaccines and the associated platform, with H.C., P.W., and K-Y.Y. included as co-inventors. There is no restriction on the publication of data. The other authors declare that they have no competing interests.
