## [Peer Review File · Nature Communications]

An intranasal influenza virus-vectored vaccine prevents SARS-CoV-2 replication in respiratory tissues of mice and hamstersReviewer #1 (Remarks to the Author):

The authors report an improvement of a previously described influenza virus-vector SARS-CoV-2 RBD vaccine approach, providing a platform for bi-functional vaccine. Since the concept of this platform has already been published, the overall novelty of the current study was incremental. In addition, there are technical concerns about the experiments.

Major points:

1. It is important to experimentally differentiate the contributions between DAF and 4N in the enhanced antibody response.
2. Glycosylation should be demonstrated for individual N mutations using Western blot. In the hamster model, day 2 viral loads are the highest. Why did the authors test day-4 viral loads?
3. How did the author determine the mRNA dose in the animal models? How could the mRNA dose be compared to the virus-vectored vaccine in terms of efficacy? Since the mRNA vaccine did not contain delta or omicron spike, you cannot compare the mRNA platform with the virus-vectored platform.
4. Do the results suggest T-cell immunity cannot efficiently cross protect other variants? How do the your results relate to human T-cell results?
5. Why was the T-cell experiment performed after 2 doses?

Minor points:

1. An explanation of how DAF allows cell membrane expression should be added.
2. The N-glycosylation 4N mutations should be more clearly explained and illustrated with neighboring amino acids to allow glycosylation.

Reviewer #2 (Remarks to the Author):

Deng et al. reports development of a potential vaccine construct blocking replication of SARS-CoV-2 by implementing a live attenuated influenza virus. The suggested construct elicits overall and neutralizing antibody response in mouse and hamster models, often surpassing those of BNT162b2 currently licensed for human vaccination. The suggested vaccine construct can be a significant measure to protect public health from the potential "twindemic" of influenza and SARS-CoV-2. The construct also demonstrated capable of blocking replication of Delta and Omicron BA.2 strains of SARS-CoV-2 in intranasally immunized animals to suggest potential further characterization of the vaccine-induced immunity for human use. However, lack of detailed immunological profiling against influenza virus and cellular immunity against SARS-CoV-2 based on currently employed mRNA vaccine constructs require further validation of the suggested vaccine candidate.

Major comments

1. Figure 1B, C: the authors transfected MDCK (dog) cell line to test for an increase of the RBD expression via DAF and 4N mutation. However, the authors later employ mouse and hamster models to characterize vaccine-induced immunity. The authors should test for the increase of the RBD expression using mouse and/or hamster cell lines to better replicate animal models.
2. Figures 1, 2, 5: Several experiments are missing mock (negative control) immunogen groups, such as BNT162b2.
3. Figure 2A: The authors should include BNT162b2 as a control, especially because they tested mouse IgG level against wild-type RBD.
4. Figure 2E: The authors need to describe potential reasons attributing to BNT162b2 (encoding wild-type) inducing lower hamster IgG level than Omi4N-DAF.
5. Figure 2E: The numbers of mice written in the legend do not match figure 2E. The legend states that Delta4N-DAF has six mice, Beta4N-DAF has four mice, Omi4N-DAF has six mice, and BNT162b2 has six mice. However, Beta4N-DAF has six dots, Omi4N-DAF has seven dots and BNT162b2 has five dots in the figure.

6. **Figure 5D: When hamsters were immunized with Omi4N*2 and subsequently challenged, they significantly lost their weight. It is concerning that it suggests a potential side effect of the vaccine construct.**
7. **Figure 6: More detailed immunological profiling after immunization with DeINS1-RBD constructs are needed to characterize protection against influenza virus.**
8. **The authors need further elaboration on the pfu (titer) used to immunize animal models based on titers for potential human immunization. Also, titer of mouse-adapted Omicron-MA is not described in the figure 4 legend.**
9. **The authors detailed information on dosage of BNT162b2 used for each animal model.**
10. **The authors need to include references comparing intranasal vs intramuscular administration of same vaccine candidate (e.g. SARS-CoV-2).**
11. **BNT162b2 encodes a full-length SARS-CoV-2 Spike protein instead of only RBD. Therefore, BNT162b1 serves a better control against the suggested DeINS1-RBD4N-DAF construct.**
12. **Lines 106-107: the authors state that "inclusion of DAF significantly increased total anti-RBD and neutralizing antibody levels in mice (Figure 1E)." However, there is no statistical analysis across the suggested groups. The authors should include a statistical analysis.**

Minor comments

1. **Revise typos and grammatical errors (e.g. line 75)**
2. **Line 77: Hyperlink is not functional.**
3. **Figure 2: color code labeling for the columns is confusing.**
4. **Figure 3: construct of CA4-DeINS1-RBD4N-DAF and HK68-DeINS1-RBD4N-DAF are confusing. The authors need to further elaborate on the acronyms and justifications on choosing the viral strains.**
5. **Figure 4: the authors should describe**
6. **Figure 5B-5G: labels of BNT*2, BNT*3, BNT*2+Delta4N, BNT*2+Omi4N, Delta4N*2 and Omi4N*2 are confusing.**
7. **It is recommended to rearrange supplemental figures as figure S5 is mentioned before S2 and S3.**

Reviewer #3 (Remarks to the Author):

The manuscript entitled **An intranasal influenza virus-vectored vaccine blocks SARS-CoV-2 replication in respiratory tissues of mice and hamsters by Deng et al**, describes the development of a dual use SARS-CoV-2/influenza intranasally administered vaccine which utilizes the live attenuated IAV-delINS1 platform that that authors have previously reported. Overall the manuscript is well designed and experiments performed well. In this study, the authors modify previous designs of RBD expressing delINS1 vector by adding glycosylation sites outside RBM and adding a membrane anchor, both of which significantly enhanced immunogenicity. While this manuscript provides important new modifications to the delINS1 vector vaccine system that the authors developed and is now in clinical trials, it may not address the major shortfalls of the clinical trial, namely immunogenicity in an influenza-experienced host.

General issues

- 1) **The authors use Students T tests to perform multiple comparisons when they should likely be using ANOVA. This may affect the interpretation of results and may not.**
- 2) **Does glycosylation and membrane anchoring increase immunogenicity in an influenza-experienced host**

Specific Issues

Figure 2:

For the Pfizer vaccine the authors should include the dose and number of inoculation in

the figure legend and methods section. It is unclear if the hamsters in the figure received prime boost or just prime and with what dose? If this data is showing only Pfizer priming than this should be repeated with prime/boost.

Figure 3:

- 1) For a vaccine study it is unclear why the authors are looking at an acute phase response rather than a memory response. The authors should repeat this assessment at more than 1 month post boost.
- 2) Additionally, in the lungs it is unclear if the authors are sampling circulating or tissue resident CD4 and CD8 cells. To address this, the authors could use pretreatment with CD45 IV labeling antibody (usually 3-5 minutes prior to euthanasia) to exclude circulating cells from the analysis.

Figure 4:

The authors should plaque purify and sequence and report the Omicron-MA and Gamma-MA viruses as mutations in the Spike may significantly affect results. The authors should report all mutations found.

Figure 5:

- 1) It is quite surprising that there are high Delta neutralizing titers induced by BNT*3 (5c) but no protection from weight loss and reduced protection for viral replication in lung and NT.
- 2) It is also quite surprising that the BNT*2+Delta4N had lower neutralizing titers to delta than did BNT*3, although better protection from viral load. This is not consistent with multiple published studies showing that antibodies are a good correlated of protection humans, mice, and NHPs.
- 3) Neutralization assay for BA.2 should also be reported in Fig 5c.
- 4) Again, in Fig S5, there Delta4N*2 induces high levels of antiviral protection (similar to that seen by Omi4N*2 despite Delta4N*2 inducing almost no BA.1 nAbs (Fig 5C). This makes it seem like the protection against omicron is non-specific. This needs to be addressed by the authors. Experimentally, I have not seen an animal model of SARS-CoV-2 allow for protection in the absence of neutralizing antibodies, nor have I seen such discordance between nAbs and protection.
- 5) There are relatively small numbers in the experiments in figure 5 and S5 and the experiment seems to have only been performed once. The authors should perform the experiments in figure 5 and S5 again and the Mock group should be DeINS1 vector vaccinated rather than mock to rule out non-specific effects of the vectored vaccine.

Figure S2/S3:

- 1) Histopathological analysis and scoring should be performed on all samples and reported.

Point-by-point response to reviewers' comments:

REVIEWER COMMENTS

Reviewer #1 (Remarks to the Author):

The authors report an improvement of a previously described influenza virus-vector SARS-CoV-2 RBD vaccine approach, providing a platform for bi-functional vaccine. Since the concept of this platform has already been published, the overall novelty of the current study was incremental. In addition, there are technical concerns about the experiments.

Major points:

1. It is important to experimentally differentiate the contributions between DAF and 4N in the enhanced antibody response.

Response:

We have performed experiments to differentiate the contributions of DAF and 4N in the enhanced antibody response and the results are included in the revised manuscript (Figure 1C, E and F).

2. Glycosylation should be demonstrated for individual N mutations using Western blot. In the hamster model, day 2 viral loads are the highest. Why did the authors test day-4 viral loads?

Response:

Western blots have been performed and the results for individual mutations are shown in Figure 1C.

We tested viral loads at 4 dpi because we also wanted to monitor body weight and lung pathology and so all infected hamsters were sacrificed on 4 dpi. We did check virus load at 2 dpi in the mouse model and found that DelNS1-RBD4N-DAF LAIVs were able to clear virus in lung tissues by 2 dpi (Figure 4 and Fig. S2). (Fig 4 shows Delta and Omi LAIVs, Fig S1 shows Delta4N-DAF LAIV).

3. How did the author determine the mRNA dose in the animal models? How could the mRNA dose be compared to the virus-vectored vaccine in terms of efficacy? Since the mRNA vaccine did not contain delta or omicron spike, you cannot compare the mRNA platform with the virus-vectored platform.

Response:

We used 1/6 (5 ug) of the human dose (30 ug), the same as has been used in other published studies. We agree with the reviewers' comment that the mRNA vaccine contains only prototype spike. Since the prototype mRNA vaccine is still being used in many countries, this study used it as a reference vaccine; the intention was not a head-to-head comparison between the mRNA vaccine and our DelNS1 viral vector vaccine which uses only RBD.

4. Do the results suggest T-cell immunity cannot efficiently cross protect other variants? How do the your results relate to human T-cell results?

Response:

Our results showed that DelNS1-RBD4N-DAF LAIVs (Delta and Omicron) induced both acute and memory phase T cell responses in lungs and spleens of immunized animals (Figure 3 and Fig. S2). Several human studies have shown cross reactivity of SARS-CoV-2 specific T cells (CD4⁺ and CD8⁺) after vaccination or infection. Given that T cells from DelNS1-RBD4N-DAF-immunized animals responded to a peptide pool based on the prototype SARS-CoV-2 strain, we believe that T cell immunity induced by DelNS1-RBD4N-DAF LAIVs is likely to have cross reactivity to other variants.

5. Why was the T-cell experiment performed after 2 doses?

Response:

In our preliminary experiments, we analyzed T cell response levels after the 1st and 2nd vaccination doses and found that 2 doses induces a higher and more stable CD4⁺ and CD8⁺ T cell response in the mouse model. We didn't check T cell responses after the second booster because the three dose regimens (BNT*3 or BNT*2 plus Delta4N or Omi4N) were only performed in hamsters and no hamster-specific reagents are available for analyzing T cell responses in hamsters.

Minor points:

1. An explanation of how DAF allows cell membrane expression should be added.

Response:

An explanation has been added in the revised version (page 5 lines 104-106).

2. The N-glycosylation 4N mutations should be more clearly explained and illustrated with neighboring amino acids to allow glycosylation.

Response:

We have added more detail in the revised version and neighboring amino acids are illustrated in the revised Figure (page 5 lines 114-116 & 122-125, Figure 1A).

Reviewer #2 (Remarks to the Author):

Deng et al. reports development of a potential vaccine construct blocking replication of SARS-CoV-2 by implementing a live attenuated influenza virus. The suggested construct elicits overall and neutralizing antibody response in mouse and hamster models, often surpassing those of BNT162b2 currently licensed for human vaccination. The suggested

vaccine construct can be a significant measure to protect public health from the potential “twindemic” of influenza and SARS-CoV-2. The construct also demonstrated capable of blocking replication of Delta and Omicron BA.2 strains of SARS-CoV-2 in intranasally immunized animals to suggest potential further characterization of the vaccine-induced immunity for human use. However, lack of detailed immunological profiling against influenza virus and cellular immunity against SARS-CoV-2 based on currently employed mRNA vaccine constructs require further validation of the suggested vaccine candidate.

Major comments

1. Figure 1B, C: the authors transfected MDCK (dog) cell line to test for an increase of the RBD expression via DAF and 4N mutation. However, the authors later employ mouse and hamster models to characterize vaccine-induced immunity. The authors should test for the increase of the RBD expression using mouse and/or hamster cell lines to better replicate animal models.

Response:

We have additionally tested expression of RBD in a golden hamster cell line (BHK21) and a human cell line (A549). In these cell lines, as well as MDCK cells, we confirmed that targeting for N-glycosylation (4N) enhanced expression of RBD by the DelNS1-RBD LAIV (Figure 1C).

2. Figures 1, 2, 5: Several experiments are missing mock (negative control) immunogen groups, such as BNT162b2.

Response:

Mock, DelNS1 viral vector and/or BNT162b2 controls have been included in our experiments throughout the revised version, as appropriate (Figure 1, 2, 3, 5, S3)).

3. Figure 2A: The authors should include BNT162b2 as a control, especially because they tested mouse IgG level against wild-type RBD.

Response:

We have performed additional experiments and BNT162b2 has been included as a control in the revised version (Figure 2).

4. Figure 2E: The authors need to describe potential reasons attributing to BNT162b2 (encoding wild-type) inducing lower hamster IgG level than Omi4N-DAF.

Response:

BNT162b2 induced a similar level of antibodies as Delta4N and Beta4N LAIV in mice but lower level of IgG than Omi4N-DAF in hamsters but lower levels in mice compared to Omi4N-DAF. It is possible that differences in vaccine intake by cells from different species may affect the immune response because BNT162b2 mRNA relies on transfection of host

cells to express antigen from mRNA. On the other hand, hamsters have been shown to be a susceptible host for influenza virus. We have added this explanation in the revised version (page 6 lines 158-161).

5. *Figure 2E: The numbers of mice written in the legend do not match figure 2E. The legend states that Delta4N-DAF has six mice, Beta4N-DAF has four mice, Omi4N-DAF has six mice, and BNT162b2 has six mice. However, Beta4N-DAF has six dots, Omi4N-DAF has seven dots and BNT162b2 has five dots in the figure.*

Response:

Our apologies for the mistake in the previous version. The number of animals used in Figure 2 have been checked and the description corrected in the revised version (Figure 2F).

6. *Figure 5D: When hamsters were immunized with Omi4N*2 and subsequently challenged, they significantly lost their weight. It is concerning that it suggests a potential side effect of the vaccine construct.*

Response:

In all our experiments, mice or hamsters immunized with DeNS1-RBD LAIVs containing Omi4N or other Delta or Beta RBDs have not shown any loss of body weight or disease symptoms in response to vaccination alone. We reason that the decrease of bodyweight on day 1-2 post infection is most likely due to the immune response countering the inoculation of virus in animals.

7. *Figure 6: More detailed immunological profiling after immunization with DelNS1-RBD constructs are needed to characterize protection against influenza virus.*

Response:

We have performed additional experiments to characterize influenza-specific immune responses in animals immunized with DelNS1-RBD constructs. These data are presented in the revised version (Figure 6F and G).

8. *The authors need further elaboration on the pfu (titer) used to immunize animal models based on titers for potential human immunization. Also, titer of mouse-adapted Omicron-MA is not described in the figure 4 legend.*

Response:

In animal experiments, all animals were immunized with 2×10^6 and 5×10^6 pfu of DelNS1-RBD4N-DAF LAIV for mice and hamsters, respectively. This information has been included in the legend of the revised version (page 24, Figure 4 legend).

9. *The authors detailed information on dosage of BNT162b2 used for each animal model.*

Response:

We used 1/6 of the human dose (5ug) in our animal experiments. Dose information has been included in the revised version (page 14 line 401).

10. The authors need to include references comparing intranasal vs intramuscular administration of same vaccine candidate (e.g. SARS-CoV-2).

Response:

References have been added in the revised version (page 3 lines 59-61, References 11-15).

11. BNT162b2 encodes a full-length SARS-CoV-2 Spike protein instead of only RBD. Therefore, BNT162b1 serves a better control against the suggested DelNS1-RBD4N-DAF construct.

Response:

We agree that full-length spike would induce additional immune responses to other targets on spike and that BNT162b1 might be a better control. However, BNT162b1 has not been imported to Hong Kong and we do not have the capacity ourselves to make a similar version of this BNT mRNA vaccine that utilizes only RBD-encoding mRNA to use as a control.

12. Lines 106-107: the authors state that “inclusion of DAF significantly increased total anti-RBD and neutralizing antibody levels in mice (Figure 1E).” However, there is no statistical analysis across the suggested groups. The authors should include a statistical analysis.

Response:

We have performed statistical analysis and show that inclusion of DAF /to promote surface presentation of RBD significantly increased total anti-RBD and neutralizing antibodies after the second dose in mice (Revised Figure 1E).

Minor comments

1. Revise typos and grammatical errors (e.g. line 75)

Response:

The correction has been made in the revised version.

2. Line 77: Hyperlink is not functional.

Response:

We have double checked the hyperlink. The Phase III trial information can also be accessed by copy-pasting this hyperlink into an internet browser.

3. Figure 2: color code labeling for the columns is confusing.

Response:

We have re-made the Figure using different colors and added Figure 2A to show information on the vaccines used in each group (Figure 2).

4. Figure 3: construct of CA4-DelNS1-RBD4N-DAF and HK68-DelNS1-RBD4N-DAF are confusing. The authors need to further elaborate on the acronyms and justifications on choosing the viral strains.

Response:

CA4-DelNS1 is the backbone viral vector which was derived from A/California/4/2009 (H1N1) (Wang P., et al., mBio 2019), and HK68-DelNS1 was made from this backbone, with HA and NA from A/Hong Kong/1/1968 (H3N2). We have added more details to descriptions of each construct in the revised version (page 14 lines 374-376).

5. Figure 4: the authors should describe

Response:

We have double checked Figure 4 and provided a more detailed description.

*6. Figure 5B-5G: labels of BNT*2, BNT*3, BNT*2+Delta4N, BNT*2+Omi4N, Delta4N*2 and Omi4N*2 are confusing.*

Response:

To help readers understand the abbreviations for vaccines used in the experiments, we have included more detailed explanations of vaccines used in all figures.

7. It is recommended to rearrange supplemental figures as figure S5 is mentioned before S2 and S3.

Response:

The supplementary figures have been rearranged accordingly in the revised version (Fig. S1-S7).

Reviewer #3 (Remarks to the Author):

The manuscript entitled An intranasal influenza virus-vectored vaccine blocks SARS-CoV-2 replication in respiratory tissues of mice and hamsters by Deng et al, describes the development of a dual use SARS-CoV-2/influenza intranasally administered vaccine which utilizes the live attenuated IAV-delNS1 platform that that authors have previously reported. Overall the manuscript is well designed and experiments performed well. In this study, the authors modify previous designs of RBD expressing delNS1 vector by adding glycosylation

sites outside RBM and adding a membrane anchor, both of which significantly enhanced immunogenicity. While this manuscript provides important new modifications to the delNS1 vector vaccine system that the authors developed and is now in clinical trials, it may not address the major shortfalls of the clinical trial, namely immunogenicity in an influenza-experienced host.

General issues

1) The authors use Students T tests to perform multiple comparisons when they should likely be using ANOVA. This may affect the interpretation of results and may not.

Response:

In the revised version, all statistical analyses have been performed using ANOVA.

2) Does glycosylation and membrane anchoring increase immunogenicity in an influenza-experienced host

Response:

To examine the immunogenicity of the DelNS1-RBD LAIV in an influenza-experienced host, we have tested the effect of prior influenza virus infection on our vaccine and found the prime boost immunization scheme used is able to induce robust SARS-CoV-2 specific antibodies and neutralizing antibodies in mice previously infected with influenza strains matching the vaccines (Fig. S3).

Specific Issues

Figure 2:

For the Pfizer vaccine the authors should include the dose and number of inoculation in the figure legend and methods section. It is unclear if the hamsters in the figure received prime boost or just prime and with what dose? If this data is showing only Pfizer priming than this should be repeated with prime/boost.

Response:

The dose and number of inoculations of BNT162b2 mRNA vaccine has been included in the figure legends and methods sections in the revised version (Page 15, line 412-414, and figure legends).

Figure 3:

1) For a vaccine study it is unclear why the authors are looking at an acute phase response rather than a memory response. The authors should repeat this assessment at more than 1 month post boost.

Response:

We have performed animal experiments to assess the memory response. Memory phase T cell responses were estimated at 9 weeks after the initial prime vaccination (5 weeks post-boost). Results are included in the revised version (Fig. S1).

2) Additionally, in the lungs it unclear if the authors are sampling circulating or tissue resident CD4 and CD8 cells. To address this, the authors could use pretreatment with CD45 IV labeling antibody (usually 3-5 minutes prior to euthanasia) to exclude circulating cells from the analysis.

Response:

We agree with the suggestion of the reviewer and have performed pretreatment with CD45 specific antibody before sacrifice of vaccine immunized animals. The results indicated that intranasal vaccination is able to induce memory T cell responses (Fig. S1).

Figure 4:

The authors should plaque purify and sequence and report the Omicron-MA and Gamma-MA viruses as mutations in the Spike may significantly affect results. The authors should report all mutations found.

Response:

The Omicron-MA and Gamma-MA viruses have been sequenced and the sequences submitted to GISAID. We have indicated mutations in the revised version of the manuscript (Supplementary Table 1).

Figure 5:

*1) It is quite surprising that there are high Delta neutralizing titers induced by BNT*3 (5c) but no protection from weight loss and reduced protection for viral replication in lung and NT.*

Response:

We think that this is the same phenomenon whereby vaccinated people still experience upper respiratory symptoms after infection. We reason that for the first few days after infection, animals will be eating less, leading to body weight loss. Recall of immunity may also take a few days and that may lead to lung pathology in infected animals. In intranasally vaccinated animals, virus was blocked or replication inhibited in the upper respiratory tract.

*2) It is also quite surprising that the BNT*2+Delta4N had lower neutralizing titers to delta than did BNT*3, although better protection from viral load. This is not consistent with multiple published studies showing that antibodies are a good correlated of protection humans, mice, and NHPs.*

Response:

In our experiments hamsters given two doses of BNT162b2 mRNA vaccine plus one booster of intranasal Delta4N (BNT*2+Delta4N) or three doses of BNT162b2 mRNA vaccine

(BNT*3) showed similar body weight loss and no significant difference in viral loads in the lungs and nasal turbinates after challenge with Delta virus (Fig. S4D, E). Published studies have focused on symptomatic disease and hospitalization, rarely estimating virus load in the airway. It is known that Delta has also gained a certain degree of immune evasion compared to the prototype virus. We agree that any animal model has limitations and that the most reliable way to evaluate a vaccine will be to test it in human trials.

3) Neutralization assay for BA.2 should also be reported in Fig 5c.

Response:

Neutralization assay results for BA.2 have been included in the revised version (Fig. S3C).

*4) Again, in Fig S5, there Delta4N*2 induces high levels of antiviral protection (similar to that seen by Omi4N*2 despite Delta4N*2 inducing almost no BA.1 nAbs (Fig 5C). This makes it seem like the protection against omicron is non-specific. This needs to be addressed by the authors. Experimentally. I have not seen an animal model of SARS-CoV-2 allow for protection in the absence of neutralizing antibodies, nor have I seen such discordance between nAbs and protection.*

Response:

In our experiment, Delta4N*2 induced moderate levels of neutralizing antibodies to Omicron BA.1 in mice but very low levels of nAb in hamsters (Figure 2C-D and 2G-H). It is possible that the level of nAb required for antiviral activity is relatively low. Another possibility is the effect of T cells, which may possess more cross protection against different variants. We showed that both Delta4N and Omi4N induce T cells reactive to epitopes derived from prototype SARS-CoV-2 virus (Figure 3).

5) There are relatively small numbers in the experiments in figure 5 and S5 and the experiment seems to have only been performed once. The authors should perform the experiments in figure 5 and S5 again and the Mock group should be DelNS1 vector vaccinated rather than mock to rule out non-specific effects of the vectored vaccine.

Response:

We have taken the suggestion of the reviewer and performed some of the animal experiments with the inclusion of a DelNS1 vector LAIV control group to verify the results. Results for vaccine protection in hamsters have been re-arranged in the revised version and demonstrate that protection is induced by DelNS1-RBD and not by non-specific activity induced by the viral vector (Figure 2, Figure 5 and Fig. S2).

Figure S2/S3:

1) Histopathological analysis and scoring should be performed on all samples and reported.

Response:

We have performed histopathological scoring and included these results in the revised version (Fig. S4 and Fig.S5).

Reviewer #1 (Remarks to the Author):

The authors have addressed this reviewer's specific comments. The novelty issue remains. This reviewer will leave it to the Editor for the final call.

Reviewer #2 (Remarks to the Author):

Deng et al. addressed most comments from the previous review. They provided more detailed explanations and clarified to support their claims.

Minor comment:

1. The authors placed emphasis on Beta, Delta and Omicron strains of SARS-CoV-2 throughout the manuscript. However, they used Gamms train of SARS-CoV-2 for the mouse-adapted model study. An explanation will help clarify the discrepancy.
2. Line 158: the authors state few cross-neutralization towards Omicron BA.1 and BA.2. It is recommended to elaborate on this sentence whether live virus or pseudovirus was used in the neutralization assay.
3. Figure 4: an explanation of contradictory results of figure4 against figure2C and 2D where Delta-RBD vaccine showed little cross-variant neutralizing activity in mice is recommended.

Reviewer #3 (Remarks to the Author):

The authors have excellently addressed my concerns

Point-by-point response to reviewers' comments:

REVIEWER COMMENTS

Reviewer #1:

The authors have addressed this reviewer's specific comments. The novelty issue remains. This reviewer will leave it to the Editor for the final call.

Response: Authors appreciate reviewer's effort and critical comments. Next-generation vaccines which can induce mucosal immunity in the upper respiratory to prevent or reduce infections caused by highly transmissible variants of SARS-CoV-2 are urgently needed. An early version of our influenza-based DelNS1-RBD LAIV intranasal vaccine for COVID-19 has recently been approved for emergency use in humans in China. This study provided strategies to further enhance immunogenicity of this vaccine for preventing SARS-CoV-2 infection and to make a bivalent vaccine for both COVID-19 and influenza.

Reviewer #2:

Deng et al. addressed most comments from the previous review. They provided more detailed explanations and clarified to support their claims.

Response: Authors appreciate reviewer's effort and critical comments.

Minor comment:

1. The authors placed emphasis on Beta, Delta and Omicron strains of SARS-CoV-2 throughout the manuscript. However, they used Gammas train of SARS-CoV-2 for the mouse-adapted model study. An explanation will help clarify the discrepancy.

Response: To facilitate experiment using mouse model, we made mouse adapted SARS-CoV-2 from Gamma and Omicron variants which contain N501Y mutation in the RBD region of spike and can be easily adapted to infect mice. Other early variants, Beta and Delta, do not contain N501Y and are more difficult to be adapted to infect mice. We showed Delta4N-DAF is effectively prevent Gamma and Delta variant infections in mice and hamster respectively (Figure 5 & Fig. S1).

2. Line 158: the authors state few cross-neutralization towards Omicron BA.1 and BA.2. It is recommended to elaborate on this sentence whether live virus or pseudovirus was used in the neutralization assay.

Response: Both pseuvirus and live virus were used in our experiments. We have modified the text to address this point in the final revision.

3. Figure 4: an explanation of contradictory results of figure4 against figure2C and 2D where Delta-RBD vaccine showed little cross-variant neutralizing activity in mice is recommended.

Response: While Delta-RBD vaccine induce lower levels of cross-variant neutralizing antibodies to Omicron BA.1 and BA.2 than Omicron-RBD (Figure 2C & D), virus challenge experiment showed that vaccination with either Delta or Omicron DelNS1-RBD4N-DAF LAIVs protected them from SARS-CoV-2 infection. We postulate that other arms of immunity including specific tissue resident T cells and mucosal IgA induce by our intranasal vaccine may act together to block and clear the challenged virus.

Reviewer #3:

The authors have excellently addressed my concerns.

Response: Authors appreciate reviewer's effort and critical comments.